# SNAKE: Shape-aware Neural 3D Keypoint Field

**Chengliang Zhong**[1,2,3]    **Peixing You**[3]    **Xiaoxue Chen**[3]    **Hao Zhao**[4,5]    **Fuchun Sun**[2]*
**Guyue Zhou**[3]    **Xiaodong Mu**[1]    **Chuang Gan**[6]    **Wenbing Huang**[7,8]
[1]Xi'an Research Institute of High-Tech    [2]THUAI, Tsinghua University
[3]AIR, Tsinghua University    [4] Peking University    [5]Intel Labs    [6]MIT
[7]Gaoling School of Artificial Intelligence, Renmin University of China
[8]Beijing Key Laboratory of Big Data Management and Analysis Methods
zhongcl19@mails.tsinghua.edu.cn; hao.zhao@intel.com;
fcsun@tsinghua.edu.cn

## Abstract

Detecting 3D keypoints from point clouds is important for shape reconstruction, while this work investigates the dual question: can shape reconstruction benefit 3D keypoint detection? Existing methods either seek salient features according to statistics of different orders or learn to predict keypoints that are invariant to transformation. Nevertheless, the idea of incorporating shape reconstruction into 3D keypoint detection is under-explored. We argue that this is restricted by former problem formulations. To this end, a novel unsupervised paradigm named SNAKE is proposed, which is short for **s**hape-aware **n**eural 3D **ke**ypoint field. Similar to recent coordinate-based radiance or distance field, our network takes 3D coordinates as inputs and predicts implicit shape indicators and keypoint saliency simultaneously, thus naturally entangling 3D keypoint detection and shape reconstruction. We achieve superior performance on various public benchmarks, including standalone object datasets ModelNet40, KeypointNet, SMPL meshes and scene-level datasets 3DMatch and Redwood. Intrinsic shape awareness brings several advantages as follows. (1) SNAKE generates 3D keypoints consistent with human semantic annotation, even without such supervision. (2) SNAKE outperforms counterparts in terms of repeatability, especially when the input point clouds are down-sampled. (3) the generated keypoints allow accurate geometric registration, notably in a zero-shot setting. Codes are available at https://github.com/zhongcl-thu/SNAKE.

## 1   Introduction

2D sparse keypoints play a vital role in reconstruction [32], recognition [22] and pose estimation [43], with scale invariant feature transform (SIFT) [19] being arguably the most important pre-Deep Learning (DL) computer vision algorithm. Altough dense alignment using photometric or featuremetric losses is also successful in various domains [2, 36, 8], sparse keypoints are usually preferred due to compactness in storage/computation and robustness to illumination/rotation. Just like their 2D counterparts, 3D keypoints have also drawn a lot of attention from the community in both pre-DL [13, 35] and DL [15, 1, 38] literature, with various applications in reconstruction [45, 41] and recognition[26, 34].

However, detecting 3D keypoints from raw point cloud data is very challenging due to sampling sparsity. No matter how we obtain raw point clouds (e.g., through RGB-D cameras [40], stereo [4], or LIDAR [10]), they are only a discrete representation of the underlying 3D shape. This fact drives us to explore the question of *whether jointly reconstructing underlying 3D shapes helps 3D*

---

*Corresponding author: Fuchun Sun.

36th Conference on Neural Information Processing Systems (NeurIPS 2022).

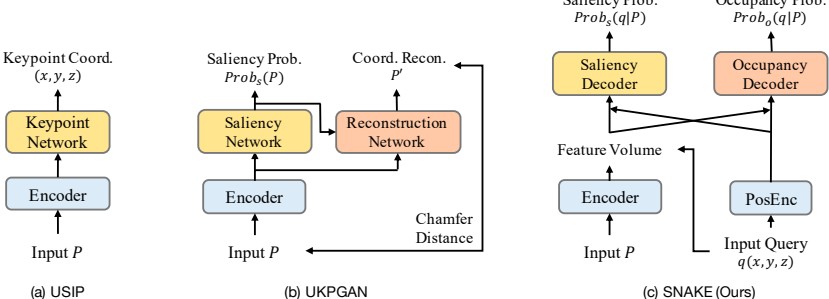

Figure 1: A comparison between existing 3D keypoint detection formulations and our newly proposed one. (a) USIP-like methods directly predict keypoint coordinates from input point clouds $P$. (b) UKPGAN-like methods predict saliency scores for $P$. Using chamfer distance, it reconstructs $P$ coordinates based on the saliency scores and latent features. (c) Our SNAKE formulation predicts saliency probabilities and shape indicators simultaneously for each *continuous* query point $q$ instead of *discrete* point clouds $P$. Sub-networks used for keypoint detection and reconstruction are shown in yellow and red, although they have different formulations. Here, the occupied points are those on the input surface.

*keypoint detection*. To our knowledge, former methods have seldom visited this idea. Traditional 3D keypoint detection methods are built upon some forms of first-order (e.g., density in intrinsic shape signature [44]) or second-order (e.g., curvature in mesh saliency [14]) statistics, including sophisticated reformulation like heat diffusion [33]. Modern learning-based methods rely upon the idea of consistency under geometric transformations, which can be imposed on either coordinate like USIP [15] or saliency value like D3Feat [1]. The most related method that studies joint reconstruction and 3D keypoint detection is a recent one named UKPGAN [38], yet it reconstructs input point cloud coordinates using an auxiliary decoder instead of the underlying shape manifold.

Why is this promising idea under-explored in the literature? We argue the reason is that former problem formulations are not naturally applicable for reconstructing the underlying shape surface. Existing paradigms are conceptually illustrated in Fig. 1. USIP-like methods directly output keypoint coordinates while UKPGAN-like methods generate saliency values for input point clouds. In both cases, the representations are based upon *discrete* point clouds. By contrast, we reformulate the problem using coordinate-based networks, as inspired by the recent success of neural radiance fields [21, 17, 29] and neural distance fields [23, 31]. As shown in Fig. 1-c, our model predicts a keypoint saliency value for each *continuous* input query point coordinate $q(x, y, z)$.

A direct advantage of this new paradigm is the possibility of tightly entangling shape reconstruction and 3D keypoint detection. As shown in Fig. 1-c, besides the keypoint saliency decoder, we attach a parallel shape indicator decoder that predicts whether the query point $q$ is occupied. The input to decoders is feature embedding generated by trilinearly sampling representations conditioned on input point clouds $P$. Imagine a feature embedding at the wing tip of an airplane, if it can be used to reconstruct the sharp curvature of the wing tip, it can be naturally detected as a keypoint with high repeatability. As such, our method is named as **s**hape-aware **n**eural 3D **ke**ypoint field, or SNAKE.

Shape awareness, as the core feature of our new formulation, brings several advantages. (1) High repeatability. Repeatability is the most important metric for keypoint detection, *i.e.*, an algorithm should detect the same keypoint locations in two-view point clouds. If the feature embedding can successfully reconstruct the same chair junction from two-view point clouds, they are expected to generate similar saliency scores. (2) Robustness to down-sampling. When input point clouds are sparse, UKPGAN-like frameworks can only achieve reconstruction up to the density of inputs. In contrast, our SNAKE formulation can naturally reconstruct the underlying surface up to any resolution because it exploits coordinate-based networks. (3) Semantic consistency. SNAKE reconstructs the shape across instances of the same category, thus naturally encouraging semantic consistency although no semantic annotation is used. For example, intermediate representations need to be similar for successfully reconstructing different human bodies because human shapes are intrinsically similar.

To summarize, this study has the following two contributions:

- We propose a new network for joint surface reconstruction and 3D keypoint detection based upon implicit neural representations. During training, we develop several self-supervised losses that exploit the mutual relationship between two decoders. During testing, we design a gradient-based optimization strategy for maximizing the saliency of keypoints.
- Via extensive quantitative and qualitative evaluations on standalone object datasets Model-Net40, KeypointNet, SMPL meshes, and scene-level datasets 3DMatch and Redwood, we demonstrate that our shape-aware formulation achieves state-of-the-art performance under three settings: (1) semantic consistency; (2) repeatability; (3) geometric registration.

## 2   Related Work

**3D Keypoint Detector** As discussed in the introduction, 3D keypoint detection methods can be mainly categorized into hand-crafted and learning-based. Popular hand-crafted approaches [44, 30, 28] employ local geometric statistics to generate keypoints. These methods usually fail to detect consistent keypoints due to the lack of global context, especially under real-world disturbances, such as density variations and noise. USIP [15] is a pioneering learning-based 3D keypoint detector that outperforms traditional methods by a large margin. However, the detected keypoints are not semantically salient, and the number of keypoints is fixed. Fernandez et al. [9] exploit the symmetry prior to generate semantically consistent keypoints. But this method is category-specific, limiting the generalization to unseen categories and scenes. Recently, UKPGAN [38] makes use of reconstruction to find semantics-aware 3D keypoints. Yet, it recovers explicit coordinates instead of implicit shape indicators. As shown in Fig. 1, different from these explicit keypoint detection methods, we propose a new detection framework using implicit neural fields, which naturally incorporates shape reconstruction.

**Implicit Neural Representation** Our method exploits implicit neural representations to parameterize a continuous 3D keypoint field, which is inspired by recent studies of neural radiance fields [17, 21, 29] and neural distance fields [23, 31, 16, 42]. Unlike explicit 3D representations such as point clouds, voxels, or meshes, implicit neural functions can decode shapes continuously and learn complex shape topologies. To obtain fine geometry, ConvONet [24] proposes to use volumetric embeddings to get local instead of global features [20] of the input. Recently, similar local geometry preserving networks show a great success for the grasp pose generation [12] and articulated model estimation [11]. They utilize the synergies between their main tasks and 3D reconstruction using shared local representations and implicit functions. Unlike [11, 12] that learn geometry as an auxiliary task, our novel losses tightly couple surface occupancy and keypoint saliency estimates.

## 3   Method

This section presents SNAKE, a shape-aware implicit network for 3D keypoint detection. SNAKE conditions two implicit decoders (for shape and keypoint saliency) on shared volumetric feature embeddings, which is shown in Fig. 2-framework. To encourage repeatable, uniformly scattered, and sparse keypoints, we employ several self-supervised loss functions which entangle the predicted surface occupancy and keypoint saliency, as depicted in the middle panel of Fig. 2. During inference, query points with high saliency are further refined by gradient-based optimization since the implicit keypoint field is continuous and differentiable, which is displayed in Fig. 2-inference.

### 3.1   Network Architecture

**Point Cloud Encoder** As fine geometry is essential to local keypoint detection, we adopt the ConvONets [24], which can obtain local details and scale to large scenes, as the point cloud encoder denoted $f_{\theta_{en}}$ for SNAKE. Given an input point cloud $P \in \mathbb{R}^{N \times 3}$, our encoder firstly processes it with the PointNet++ [25] (or alternatives like [46]) to get a feature embedding $Z \in \mathbb{R}^{N \times C_1}$, where $N$ and $C_1$ are respectively the number of points and the dimension of the features. Then, these features are projected and aggregated into structured volume $Z' \in \mathbb{R}^{C_1 \times H \times W \times D}$, where $H$, $W$ and $D$ are the number of voxels in three orthogonal axes. The volumetric embeddings serve as input to the 3D UNet [6] to further integrate local and global information, resulting in the output $G \in \mathbb{R}^{C_2 \times H \times W \times D}$, where $C_2$ is the output feature dimension. More details can be found in the Appendix.

**Shape Implicit Decoder** As shown in the top panel of Fig. 2, each point $q \in \mathbb{R}^3$ from a query set $Q$ is encoded into a $C_e$-dimensional vector $q_e$ via a multi-layer perceptron that is denoted the positional

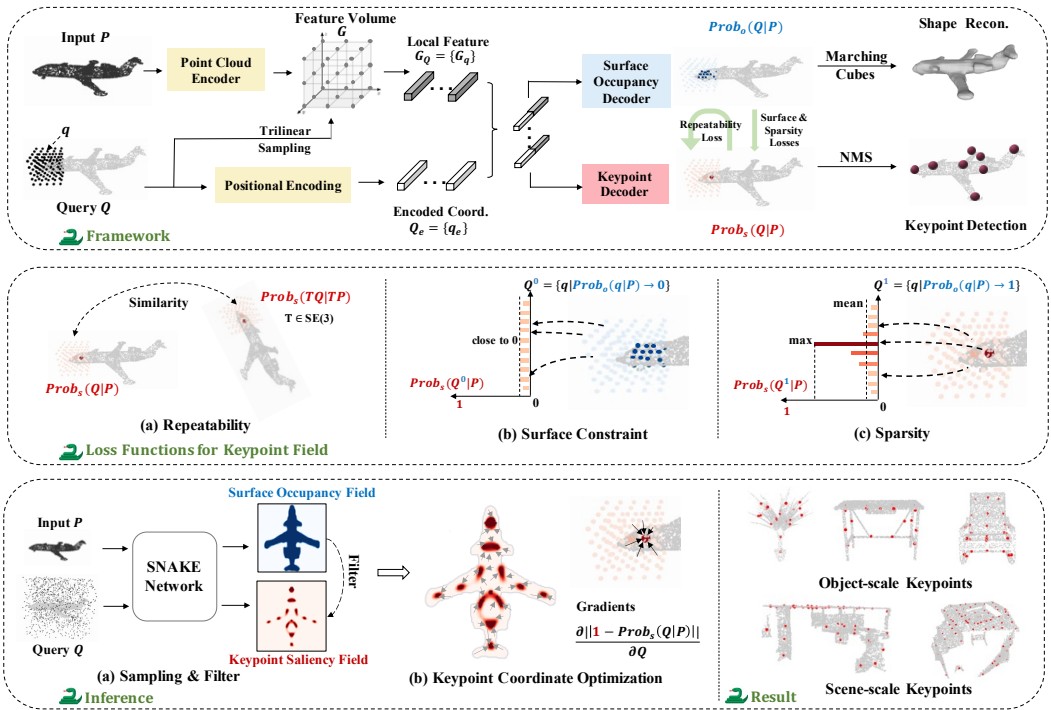

Figure 2: **Framework:** We use an implicit network to decode the surface occupancy and keypoint saliency probability simultaneously. Green arrows indicate the mutual relationships between the geometry and saliency field. Through marching cubes and non-maximum suppression (NMS), it could respectively recover the shape and detect keypoints from the input. **Loss functions for keypoint filed:** Three loss functions try to make the generated keypoint repeatable, located on the underlying surface, and sparse. **Inference:** We design a gradient-based optimization method to extract keypoints from the saliency field. **Result:** The object-scale and scene-scale keypoints after inference are displayed.

encoder $f_{\theta_{pos}}$, *i.e.* $q_e = f_{\theta_{pos}}(q)$. Then, the local feature $G_q$ is retrieved from the feature volume $G$ according to the coordinate of $q$ via trilinear interpolation. The generated $q_e$ and $G_q$ are concatenated and mapped to the surface occupancy probability $Prob_o(q|P) \in [0, 1]$ by the occupancy decoder $f_{\theta_o}$, as given in Eq. (1). If $q$ is on the input surface, the $Prob_o(q|P)$ would be 1, otherwise be 0. In our formulation, the points inside the surface are also considered unoccupied.

$$f_{\theta_o}(q_e, G_q) \rightarrow Prob_o(q|P) \tag{1}$$

**Keypoint Implicit Decoder** Most of the process here is the same as in shape implicit decoder, except for the last mapping function. The goal of keypoint implicit decoder $f_{\theta_s}$ is to estimate the saliency of the query point $q$ conditioned on input points $P$, which is denoted as $Prob_s(q|P) \in [0, 1]$ and formulated by:

$$f_{\theta_s}(q_e, G_q) \rightarrow Prob_s(q|P). \tag{2}$$

Here, saliency of the query point $q$ is the likelihood that it is a keypoint.

## 3.2 Implicit Field Training

The implicit field is jointly optimized for surface occupancy and saliency estimation by several self-supervised losses. In contrast to former arts [12, 11] with a similar architecture that learn multiple tasks separately, we leverage the geometry knowledge from shape field to enhance the performance of keypoint field, as shown in the green arrows of Fig. 2. Specifically, the total loss is given by:

$$\mathcal{L} = \mathcal{L}_o + \mathcal{L}_r + \mathcal{L}_m + \mathcal{L}_s, \tag{3}$$

where $\mathcal{L}_o$ encourages the model to learn the shape from the sparse input, $\mathcal{L}_r$, $\mathcal{L}_m$ and $\mathcal{L}_s$ respectively help the predicted keypoint to be repeatable, located on the underlying surface and sparse.

**Surface Occupancy Loss** The binary cross-entropy loss $l_{\text{BCE}}$ between the predicted surface occupancy $Prob_o(q|P)$ and the ground-truth label $Prob_o^{gt}$ is used for shape recovery. The queries $Q$ are randomly sampled from the whole volume size $H \times W \times D$. The average over all queries is as follows:

$$\mathcal{L}_o = \frac{1}{|Q|} \sum_{q \in Q} l_{\text{BCE}}\big(Prob_o(q|P), Prob_o^{gt}(q|P)\big), \tag{4}$$

where $|Q|$ is the number of queries $Q$.

**Repeatability Loss** Detecting keypoints with high repeatability is essential for downstream tasks like registration between two-view point clouds. That indicates the positions of keypoint are covariant to the rigid transformation of the input. To achieve a similar goal, 2D keypoint detection methods [27, 7, 43] enforce the similarity of corresponding local salient patches from multiple views. Inspired by them, we enforce the similarity of local overlapped saliency fields from two-view point clouds. Since the implicit field is continuous, we uniformly sample some values from a local field to represent the local saliency distribution. Specifically, as shown in the top and the middle part of Fig. 2, we build several local 3D Cartesian grids $\{Q_i\}_{i=1}^n$ with resolution of $H_l \times W_l \times D_l$ and size of $1/U$. We empirically set the resolution of $Q_i$ to be almost the same as the feature volume $G$. As non-occupied regions are uninformative, the center of $Q_i$ is randomly sampled from the input. Then, we perform random rigid transformation $T$ on the $P$ and $Q_i$ to generate $TP$ and $TQ_i$. Similar to [27], the cosine similarity, denoted as $\text{cosim}$, is exploited for the corresponding saliency grids of $Q_i$ and $TQ_i$:

$$\mathcal{L}_r = 1 - \frac{1}{n} \sum_{i \in n} \text{cosim}\big(Prob_s(Q_i|P), Prob_s(TQ_i|TP)\big). \tag{5}$$

**Surface Constraint Loss** As discussed in [15], 3D keypoints are encouraged to close to the input. They propose a loss to constrain the distance between the keypoint and its nearest neighbor from the input. Yet, the generated keypoints are inconsistent when given the same input but with a different density. Thanks to the shape decoder, SNAKE can reconstruct the underlying surface of the input, which is robust to the resolution change. Hence, we use the surface occupancy probability to represent the inverse distance between the query and the input. As can be seen in Fig. 2-(surface constraint), we enforce the saliency of the query that is far from input $P$ close to 0, which is defined as

$$\mathcal{L}_m = \frac{1}{|Q|} \sum_{q \in Q} \big(1 - Prob_o(q|P)\big) \cdot Prob_s(q|P). \tag{6}$$

**Sparsity Loss** Similar to 2D keypoint detection methods [27], we design a sparsity loss to avoid the trivial solution ($Prob_s(Q|P)$=0) in Eq.( 5)( 6). As can be seen in Fig. 2, the goal is to maximize the local peakiness of the local saliency grids. As the sailency values of non-occupied points are enforced to 0 by $\mathcal{L}_m$, we only impose the sparsity loss on the points with high surface occupancy probability. Hence, we derive the sparsity loss with the help of decoded geometry by

$$\mathcal{L}_s = 1 - \frac{1}{n} \sum_{i \in n} \big( \max Prob_s(Q_i^1|P) - \text{mean} \, Prob_s(Q_i^1|P)\big), \tag{7}$$

where $Q_i^1 = \{q | q \in Q_i, Prob_o(q|P) > 1 - thr_o\}$, $thr_o \in (0, 0.5]$ is a constant, and $n$ is the number of grids. It is noted that the spatial frequency of local peakiness is dependent on the grid size $1/U$, see section 4.4. Since the network is not only required to find sparse keypoints, but also expected to recover the object shape, it would generate high saliency at the critical parts of the input, like joint points of a desk and corners of a house, as shown in the Fig. 2-result.

## 3.3 Explicit Keypoint Extraction

The query point $q$ whose saliency is above a predefined threshold $thr_s \in (0, 1)$ would be selected as a keypoint at the inference stage. Although SNAKE can obtain the saliency of any query point, a higher resolution query set results in a high computational cost. Hence, as shown in Fig. 2-inference, we build a relatively low-resolution query sets $Q_{\text{infer}}$ which are evenly distributed in the input space and further refine the coordinates of $Q_{\text{infer}}$ by gradient-based optimization on this energy function:

$$E(Q_{\text{infer}}, P) = \frac{1}{|Q_{\text{infer}}|} \sum_{q \in Q_{\text{infer}}} 1 - Prob_s(q|P). \tag{8}$$

Specifically, details of the explicit keypoint extraction algorithm are summarized in Alg. 1.

**Algorithm 1** Optimization for Explicit Keypoint Extraction
___
**Require:** $P, Q_{\text{infer}}, f_{\theta_{en}}, f_{\theta_{pos}}, f_{\theta_o}, f_{\theta_s}$. Hyper-parameters: $\lambda, J, thr_o, thr_s$.
  Get initial $Prob_o(Q_{\text{infer}}|P)$ according to Eq.( 1).
  Filter to get new query set $Q_{\text{infer}'} = \{q|q \in Q_{\text{infer}}, Prob_o(q|P) > 1 - thr_o\}$.
  **for** 1 to $J$ **do**
    Evaluate energy function $E(Q_{\text{infer}'}, P)$.
    Update coordinates with gradient descent: $Q_{\text{infer}'} = Q_{\text{infer}'} - \lambda \nabla_{Q_{\text{infer}'}} E(Q_{\text{infer}'}, P)$.
  **end for**
  Sample final keypoints $Q_k = \{q|q \in Q_{\text{infer}'}, Prob_s(q|P) > thr_s\}$.
___

## 4   Experiment

In this section, we evaluate SNAKE under three settings. First, we compare keypoint semantic consistency across **different instances** of the same category, using both rigid and deformable objects. Next, keypoint repeatability of the **same instance** under disturbances such as SE(3) transformation, noise and downsample is evaluated. Finally, we inspect the point cloud registration task on the 3DMatch benchmark, notably in a zero-shot generalization setting. Besides, an ablation study is done to verify the effect of each design choice in SNAKE. The implementation details and hyper-parameters for SNAKE in three settings can be found in the Appendix.

### 4.1   Semantic Consistency

**Datasets** The KeypointNet [39] dataset and meshes generated with the SMPL model [18] are utilized. KeypointNet has numerous human-annotated 3D keypoints for 16 object categories from ShapeNet [3]. The training set covers all categories that contain 5500 instances. Following [38], we evaluate 630 unseen instances from airplanes, chairs, and tables. SMPL is a skinned vertex-based deformable model that accurately captures body shape variations in natural human poses. We use the same strategy in [38] to generate both training and testing data.

**Metric** Mean Intersection over Union (mIoU) is adopted to show whether the keypoints across intra-class instances have the same semantics or not. For KeypointNet, a predicted keypoint is considered the same as a human-annotated semantic point if the geodesic distance between them is under some threshold. Due to the lack of human-labeled keypoints on SMPL, we compare the keypoint consistency in a pair of human models. A keypoint in the first model is regarded semantically consistent if the distance between its corresponding point and the nearest keypoint in the second model is below some threshold.

**Evaluation and Results**   We compare SNAKE with random detection, hand-crafted detectors: ISS [44], Harris-3D [30] and SIFT-3D [28], and DL-based unsupervised detectors: USIP [15] and UKPGAN [38]. As USIP has not performed semantic consistency evaluations, we train the model with the code they provided. We follow the same protocols in [38] to filter the keypoints via NMS with a Euclidean radius of 0.1. Quantitative results are provided in Fig. 5-(a,e).

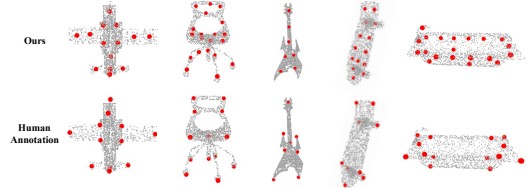

Figure 3: Comparison with human annotations on KeypointNet [39] dataset.

SNAKE obtains higher mIoU than other methods under most thresholds on KeypointNet and SMPL. Qualitative results in Fig. 3 show our keypoints make good alignment with human annotations. Fig. 4 provides qualitative comparisons of semantically consistent keypoints on rigid and deformable objects. Owing to entangling shape reconstruction and keypoint detection, SNAKE can extract aligned representation for intra-class instances. Thus, our keypoints better outline the object shapes and are more semantically consistent under large shape variations. As shown in the saliency field projected slices, we can get symmetrical keypoints, although without any explicit constraint like the one used in [38]. Here, a projected slice is obtained by taking the maximum value of a given field along the projection direction.

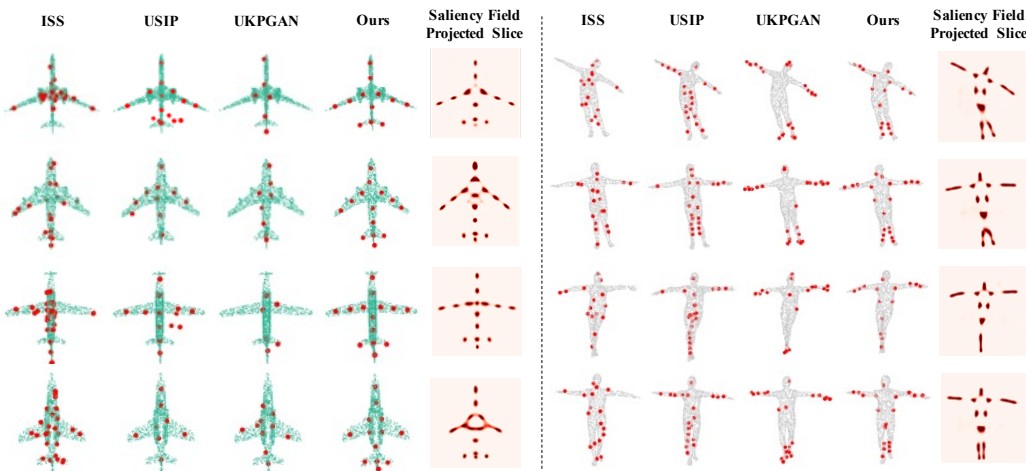

Figure 4: Semantic consistency of keypoints on rigid and deformable objects. Our keypoints are more evenly scattered on the underlying surface of objects, more symmetrical, and more semantically consistent under significant shape variations when compared to other methods. The saliency field projected slice shows that SNAKE decodes well-aligned saliency values for keypoints in different instances but with similar semantics, such as the wingtip of the airplane and the leg of the human. Here, small saliency is shown in bright red and gets darker with a larger value.

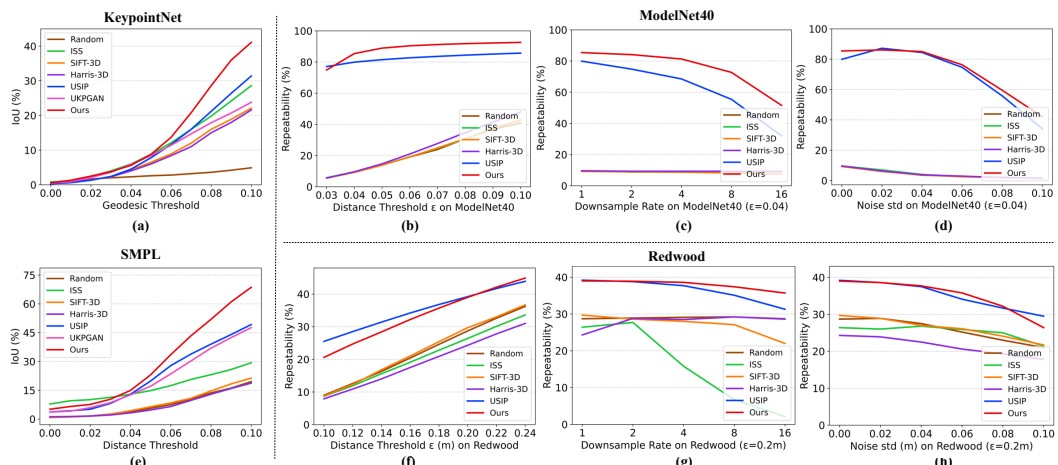

Figure 5: Quantitative results on four datasets. Keypoint semantic consistency (a)(e) on KeypointNet and SMPL. Relative repeatability for two-view point clouds with different distance threshold (b), downsample rate (c), Gaussian noise $\mathcal{N}(0, \sigma_{noise})$ (d) on ModelNet40. The results of (f)(g)(h) are tested on Redwood with the same settings in (b)(c)(d). The specific numerical results can be found in the Appendix.

## 4.2 Repeatability

**Datasets** ModelNet40 [37] is a synthetic object-level dataset that contains 12,311 pre-aligned shapes from 40 categories, such as plane, guitar, and table. We adopt the official dataset split strategy. 3DMatch [41] and Redwood [5] are RGB-D reconstruction datasets for indoor scenes. Following [15], we train the model on 3DMatch and test it on Redwood to show the generalization performance. The training set contains around 19k samples and the test set consists of 207 point clouds.

**Metric** We adopt the relative repeatability proposed in USIP [15] as the evaluation metric. Given two point clouds captured from different viewpoints, a keypoint in the first point cloud is repeatable if its

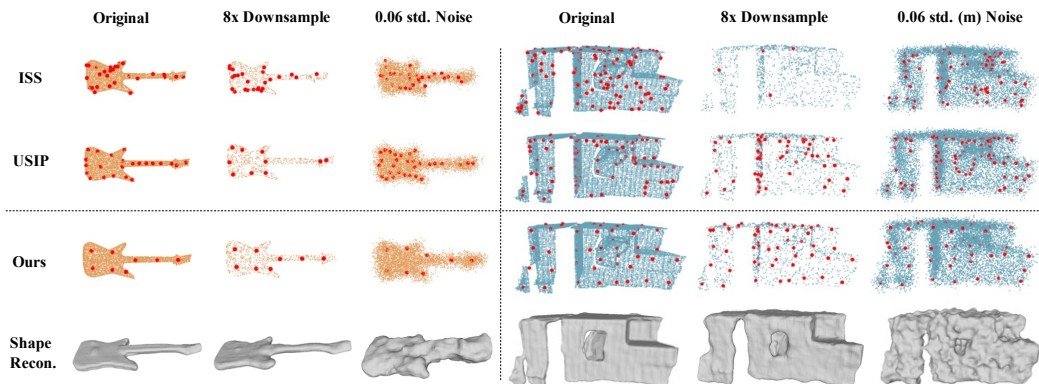

Figure 6: Visualization of keypoints under some disturbances on object-level [37] and scene-level [5] datasets compared to hand-crafted [44] and explicit representation based [15] methods. Downsample rate is 8x and the Gaussian noise scale ($\sigma$) is 0.06. The shape reconstruction via marching cubes for our occupancy field is also given. Visualization of repeatability can be found in the Appendix.

distance to the nearest keypoint in the other point cloud is below a threshold $\epsilon$. *Relative* repeatability means the number of repeatable points divided by the total number of detected keypoints.

**Evaluation and Results** Random detection, traditional methods and USIP are chosen as our baselines. Since UKPGAN does not provide pre-trained models on these two datasets, we do not report its results in Fig. 5 but make an additional comparison on KeypointNet, which is illustrated in the next paragraph. We use NMS to select the local peaky keypoints with a small radius (0.01 normalized distance on ModelNet40 and 0.04 meters on Redwood) for ours and baselines. We generate 64 keypoints in each sample and show the performance under different distance thresholds $\epsilon$, downsample rates, and Gaussian noise scales. We set a fixed $\epsilon$ of 0.04 normalized distance and 0.2 meters on the ModelNet40 and Redwood dataset when testing under the last two cases. As shown in Fig. 5-(b,f), SNAKE outperforms state-of-the-art at most distance thresholds. We do not surpass USIP on Redwood in the lower thresholds. Note that it is challenging to get higher repeatability on Redwood because the paired inputs have very small overlapping regions. Fig. 5-(c,d,g,h) show the repeatability robustness to different downsample rates (d.r.) and Gaussian noise $N(0, \sigma)$ levels. SNAKE gets the highest repeatability in most cases because the shape-aware strategy helps the model reason about the underlying shapes of the objects/scenes, which makes keypoints robust to the input variations. Fig. 6 provides visualization of object-level and scene-level keypoints of the original and disturbed inputs. SNAKE can generate more consistent keypoints than other methods under drastic input changes.

We have tried to train UKPGAN (official implementation) on ModelNet40 and 3DMatch datasets from scratch but observed divergence under default hyper-parameters. As such, we provide a new experiment to compare their repeatability on the KeypointNet dataset, on which UKPGAN provided a pre-trained model. We randomly perform SE(3) transformation on the test point clouds to generate the second view point clouds. Then, we select top-32 salient keypoints with NMS (radius=0.03) in each sample and report the keypoint repeatability under different distance thresholds $\epsilon$, downsample rates, and Gaussian noise scales. The results are summarized in Table 1, 2, which show that SNAKE achieves significant gains over UKPGAN in most cases. More discussions can be found in the Appendix.

Table 1: Relative repeatability (%) with different distance thresholds $\epsilon$ on the KeypointNet dataset.

|  | 0.03 | 0.05 | 0.07 | 0.09 | 0.10 |
|---|---|---|---|---|---|
| UKPGAN | 0.199 | 0.454 | 0.661 | 0.81 | 0.864 |
| Ours | **0.643** | **0.806** | **0.892** | **0.936** | **0.948** |

Table 2: Relative repeatability (%) when input point clouds are disturbed ($\epsilon$=0.03). Here, ori. means the original input.

|  | ori. | d.r.=4 | d.r.=8 | $\sigma$=0.02 | $\sigma$=0.03 |
|---|---|---|---|---|---|
| UKPGAN | 0.199 | 0.570 | 0.427 | 0.608 | **0.558** |
| Ours | **0.643** | **0.594** | **0.525** | **0.626** | 0.536 |

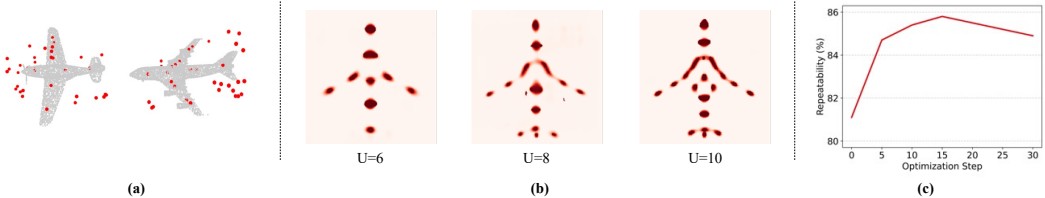

| (a) | (b) | (c) |

Figure 7: (a) SNAKE fails to predict semantically consistent keypoints without the occupancy decoder. (b) Saliency field slice with a different grid size of $(1/U)^3$. (c) The impact of the optimization step.

## 4.3 Zero-shot Point Cloud Registration

**Datasets** We follow the same protocols in [38] to train the model on KeypointNet and then directly test it on 3DMatch [41] dataset, evaluating how well two-view point clouds can be registered. The test set consists of 8 scenes which include some partially overlapped point cloud fragments and the ground truth SE(3) transformation matrices.

**Metric** To evaluate geometric registration, we need both keypoint detectors and descriptors. Thus, we combine an off-the-shelf and state-of-the-art descriptor D3Feat [1] with our and other keypoint detectors. Following [38], we compute three metrics: Feature Matching Recall, Inlier Ratio, and Registration Recall for a pair of point clouds.

**Evaluation and Results** As baselines, we choose random detection, ISS, SIFT-3D, UKPGAN, and D3Feat. Note that D3Feat is a task-specific learning-based detector trained on the 3DMatch dataset, thus not included in this zero-shot comparison. Ours and UKPGAN are trained on the synthetic object dataset KeypointNet only. The results are reported under different numbers of keypoints (*i.e.*, 2500, 1000, 500, 250, 100). The NMS with a radius of 0.05m is used for D3Feat, UKPGAN, and ours. As shown in Table 3, SNAKE outperforms other methods consistently under three metrics. For registration recall and inlier ratio, we achieve significant gains over UKPGAN and other traditional keypoint methods. Notably, when the keypoints are high in numbers, SNAKE even outperforms D3Feat which has seen the target domain. Local shape primitives like planes, corners, or curves may be shared between objects and scenes, so our shape-aware formulation allows a superior generalization from objects to scenes.

Table 3: Registration result on 3DMatch. We combine the off-the-shelf descriptor D3Feat [1] and different keypoint detectors to perform two-view point cloud registration.

| Detector | Descriptor | Feature Matching Recall (%) | | | | | Registration Recall (%) | | | | | Inlier Ratio (%) | | | | |
|---|---|---|---|---|---|---|---|---|---|---|---|---|---|---|---|---|
| | | 2500 | 1000 | 500 | 250 | 100 | 2500 | 1000 | 500 | 250 | 100 | 2500 | 1000 | 500 | 250 | 100 |
| D3Feat | D3Feat | 95.6 | 94.5 | 94.3 | 93.3 | 90.6 | 84.4 | 84.9 | 82.5 | 79.3 | 67.2 | 40.6 | 42.7 | 44.1 | 45.0 | 45.6 |
| Random | D3Feat | 95.1 | 94.5 | 92.8 | 90.0 | 81.2 | 83.0 | 80.0 | 77.0 | 65.5 | 38.8 | 38.6 | 33.6 | 28.9 | 23.6 | 17.3 |
| ISS | D3Feat | 95.2 | 94.4 | 93.4 | 90.1 | 81.0 | 83.5 | 79.2 | 76.0 | 64.3 | 37.2 | 38.2 | 33.5 | 28.8 | 23.9 | 17.4 |
| SIFT | D3Feat | 94.9 | 94.0 | 93.0 | 91.2 | 81.3 | 84.0 | 79.9 | 76.1 | 60.9 | 38.6 | 38.4 | 33.6 | 28.8 | 23.3 | 17.4 |
| UKPGAN | D3Feat | 94.7 | 94.2 | 93.5 | 92.6 | 85.9 | 82.8 | 81.4 | 77.1 | 69.7 | 47.4 | 38.8 | 35.5 | 34.0 | 33.1 | 27.7 |
| Ours | D3Feat | **95.5** | **95.0** | **94.7** | **92.9** | **89.5** | **85.1** | **83.7** | **81.2** | **74.6** | **50.9** | **41.3** | **39.0** | **37.0** | **33.5** | **30.0** |

## 4.4 Ablation Study

**Loss Function** Table 4 reports the performance w.r.t. designs of loss functions. (Row 1) If the surface occupancy decoder is removed, the surface constraint cannot be performed according to Eq.( 6), so they are removed simultaneously. Although the model could detect significantly repeatable keypoints on ModelNet40 [37], it fails to give semantically consistent keypoints on KeypointNet [39]. Fig. 7-a shows that SNAKE is unable to output symmetric and meaningful keypoints without the shape-aware technique. That indicates the repeatability could not be the only criterion for keypoint detection if an implicit formulation is adopted. (Row 2-4) Each loss function for training keypoint field is vital for keypoint detection. Note that the model gives a trivial solution (0) for the saliency field and cannot extract distinctive points when removing the sparsity loss.

**Grid Size and Volumetric Resolution** The grid size $1/U$ controls the number of keypoints because $\mathcal{L}_s$ enforces the model to predict a single local maxima per grid of size $(1/U)^3$. Fig. 7-b shows

Table 4: Ablations for the designs of loss function. occ. = occupancy, sur. = surface, rep. = repeatability, spa. = sparsity and rr. = relative repeatability.

| Threshold $\epsilon$ | rr. (%) on [37] | | | mIoU (%) on [39] | | |
|---|---|---|---|---|---|---|
| | 0.04 | 0.05 | 0.06 | 0.08 | 0.09 | 0.1 |
| w/o occ. & sur. | **0.92** | **0.94** | **0.95** | 0.22 | 0.25 | 0.28 |
| w/o sur. | 0.28 | 0.36 | 0.42 | **0.31** | 0.35 | 0.39 |
| w/o rep. | 0.22 | 0.28 | 0.34 | 0.30 | 0.35 | 0.39 |
| w/o spa. | 0 | 0 | 0 | 0 | 0 | 0 |
| w/ all | 0.85 | 0.89 | 0.90 | 0.30 | **0.37** | **0.42** |

Table 5: Impact of different local grid size used in the $\mathcal{L}_o$ and $\mathcal{L}_s$ on ModelNet40.

| $U$ | 4 | 6 | 8 | 10 |
|---|---|---|---|---|
| rr. (%) ($\epsilon$=0.04) | 0.79 | **0.85** | 0.79 | 0.77 |

Table 6: Impact of different global volumetric resolution on ModelNet40.

| $H(=W=D)$ | 32 | 48 | 64 | 80 |
|---|---|---|---|---|
| rr. (%) ($\epsilon$=0.04) | 0.62 | 0.79 | **0.85** | 0.78 |

different saliency field slices obtained from the same input with various $1/U$. When $U$ is small, SNAKE outputs fewer salient responses, and more for larger values of $U$. We also give the relative repeatability results on ModelNet40 under distance threshold $\epsilon = 0.04$ in Table 5, indicating that $U = 6$ gives the best results. From Table 6, we can see that higher resolution improves performance. However, the performance drops when it reaches the resolution of 80. The potential reason is as such: the number of queries in a single grid increases when the resolution becomes higher, as mentioned in 3.2. In this case, finer details make the input to cosine similarity too long and contain spurious values.

**Optimization Step and Learning Rate** Fig. 7-c shows the importance of optimization (see Alg. 1) for refining keypoint coordinates on the ModelNet40 dataset. It is noted that too many optimization steps will not bring more gains but increase the computational overhead. In this paper, we set the number of update steps to 10. The learning rate for optimization is also key to the final result. When the learning rate is set to 0.1, 0.01, 0.001 and 0.0001, the relative repeatability (%) on ModelNet40 dataset with the same experimental settings as Table 6 are 0.002, 0.622, 0.854 and 0.826, respectively. In addition, the comparison of computation cost of baselines and ours can be found in the Appendix.

## 5 Conclusion and Discussion

We propose SNAKE, a method for 3D keypoint detection based on implicit neural representations. Extensive evaluations show our keypoints are semantically consistent, repeatable, robust to downsample, and generalizable to unseen scenarios. **Limitations.** The optimization for keypoint extraction during inference requires considerable computational cost and time, which may not be applicable for use in scenarios that require real-time keypoint detection. **Negative Social Impact.** The industry may use the method for pose estimation in autonomous robots. Since our method is not perfect, it may lead to wrong decision making and potential human injury.

## 6 Acknowledgments

This research is jointly supported by following projects: the Scientific Innovation 2030 Major Project for New Generation of AI under Grant No.2020AAA0107300, Ministry of Science and Technology of the People's Republic of China; the Key Field R&D Program of Guangdong Province (No.2021B0101410002); Sino-German Collaborative Research Project Crossmodal Learning (NSFC 61621136008/DFG SFB/TRR169); the National Natural Science Foundation of China (No.62006137); Beijing Outstanding Young Scientist Program (No.BJJWZYJH012019100020098). We would like to thank Pengfei Li for discussions about implicit field learning. We would also like to thank the anonymous reviewers for their insightful comments.

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
