# OpenReview forum: "SNAKE: Shape-aware Neural 3D Keypoint Field"
_NeurIPS.cc/2022/Conference — NeurIPS 2022 Accept_

### Official Review · Reviewer_xHwu · 2022-07-06

**Rating:** 3
**Confidence:** 5
**Soundness:** 2 fair
**Presentation:** 2 fair
**Contribution:** 1 poor

**Summary:**

The authors introduced a method to detect keypoints from point clouds. They leverage a deep learning model to learn an implicit function which is a mapping to provide each location a probability of being a keypoint. To make the method shape-aware, the authors use deep learning models to learn the kepoint field along with the learning of an occupancy field to explore whether the shape information can improve the keypoint detection. The contribution lies in the way of combining the learning of implicit fields and keypoint fields. The authors evaluate the effectiveness by comparing it with the latest methods under the widely used benchmark.

**Questions:**

1. First of all, I do not think it makes any sense to learn the probability of keypoints as a field. Keypoints should be located on surfaces, they are not floating in the space. Specifically, the authors aim to detect keypoints from input point clouds, the keypoints to be determined should be some of the input points. It should be much easier to learn the probability of keypoints among the discrete input points than in the continuous whole space.

2. The motivation of improving the performance with surface reconstruction does not make any sense either. Surface reconstruction is a harder task than keypoint detection, since point clouds have been given as input which provides a range as a good constraint to find solutions for kepoint detection. We already know the structure of shapes represented by point clouds and why we have to learn an implicit function to reconstruct shapes. I believe it just makes this problem more complex to get resolved.

3. The authors claim that the proposed method is a novel unsupervised method. This is also not correct, since this method requires ground truth occupancy information as a supervision.

4. The proposed method is also not novel. UKPGAN has explored the feasibility of combining shape reconstruction and keypoint detection together, although UKPGAN reconstructs the input point cloud rather than learning an implicit function to represent the same shape. The minor difference here is what representation is used to represent 3D shapes.

5. Learning occupancy field results in using an additional surface constraint. This term should not be there due to the learning of the occupancy field. Fig.2(b) may be confusing, I think the close to 0 line should be parallel to the x axis rather than y axis.

6. The term of repeatability loss is hard to understand. Why do we need a term like this? For shapes, they are well aligned, why do we need to consider the rigid transformation? Another question is about the cosine similarity, the probabilities are scalars, right? Is there any reason to use cosine similarity to evaluate the difference between two numbers?

7. The authors use occupancy probability to represent the inverse distance between the query and the input. I do not think it works here. Since occupancy probability always gets smaller from 1 to 0 when moving a query from inside to outside while the inverse distance gets smaller to larger before going across the surface, and then gets smaller after that.

8. Explicit keypoint extraction via optimization is also an operation that I do not understand. An intuitive idea of extracting keypoints is to use points of the input point clouds as queries to predict the occupancy values since keypoints can only locate on the surface. Why do we have to extract keypoints via updating the locations of randomly sampled queries in the 3D space?

9. In the visualization of saliency field slices, how to select the slices to visualize?

10. How to determine the keypoint number? I noticed that the numbers of keypoints produced by different methods are different even in the same case, such as the visual comparison in Fig.6. Why do not use the same number of points to perform the comparison？

11. The reason why I do not think the reconstruction can improve the performance of keypoint detection is the results in Table 2. The results without surface reconstruction are better than the results with surface reconstruction. I do not think this is a good support for the argument.

12. With the ground truth occupancy supervision, it is not a fair comparison with other methods.


**Limitations:**

Yes, the authors addressed the limitations and potential negative societal impact of their work.

**Strengths And Weaknesses:**

Strengths:
1. The visualization is good.
2. The paper is easy to follow.


Weaknesses:
1. The motivation is not convincing at all.
2. The experimental results cannot justify the effectiveness of the method.

---

> ### Author Response · Authors · 2022-08-02
> **Responses to Reviewer #xHwu 1/3**
>
> We would like to thank R#xHwu for the detaield comments. Here we respond to raised concerns one by one.
>
> ---
>
> **Q1**:
>
> > First of all, I do not think it makes any sense to learn the probability of keypoints as a field. Keypoints should be located on surfaces, they are not floating in the space. Specifically, the authors aim to detect keypoints from input point clouds, the keypoints to be determined should be some of the input points. It should be much easier to learn the probability of keypoints among the discrete input points than in the continuous whole space.
>
> **A1**:
>
> (1) We do agree that enforcing keypoints to lie on the surface is a reasonable choice. To this end, we have proposed two techniques: surface constraint loss during training and occupancy filtering during inference.
>
> (2) Point clouds are limited in terms of the number of points that may not contain the exact keypoints but only the points near the keypoints. Therefore, we want the detector to have the potential to predict keypoints drift away from the input points to improve keypoint localization. USIP holds the same view that it is unnecessary for keypoints to be any of the input points. Since SNAKE learns a continuous keypoint field that can be queried at any point in 3D space, it can do so. The various experimental results in the main paper also verify the effectiveness of our method. If R#xHwu insists that the keypoints must be derived from the input points, then use the query set sampled from input points and set no optimization when making inference for our network.
>
> (3) In classical 2d keypoint detection methods like SIFT[1] or SURF[2], subsequent steps for sub-pixel refinement are widely used to improve keypoint localization, which means that 2D keypoints do not necessarily lie on the input pixel grid.
>
> (4) When the test point clouds are disturbed by downsampling or noises, the methods like UKPGAN, which generates keypoints from discrete inputs, cannot maintain a consistent performance, which can be found in section C.1 in the supplementary.
>
> (5) We do not use any additional information that other methods do not have access to (see A3 for more details). Our entire training process is easy and stable.
>
> [1] Lowe, David G. "Object recognition from local scale-invariant features." Proceedings of the seventh IEEE international conference on computer vision. Vol. 2. Ieee, 1999.
>
> [2]Bay, Herbert, Tinne Tuytelaars, and Luc Van Gool. "Surf: Speeded up robust features." European conference on computer vision. Springer, Berlin, Heidelberg, 2006.
>
> ---
>
> **Q2**:
>
> > The motivation of improving the performance with surface reconstruction does not make any sense either. Surface reconstruction is a harder task than keypoint detection, since point clouds have been given as input which provides a range as a good constraint to find solutions for kepoint detection. We already know the structure of shapes represented by point clouds and why we have to learn an implicit function to reconstruct shapes. I believe it just makes this problem more complex to get resolved.
>
> **A2**:
>
> (1) Point cloud is not a complete 3D representation for 3D data because its sampling rate is always limited. Furthermore, it cannot represent topological relations. In contrast, implicit functions can represent 3D data of arbitrary topology and at arbitrary resolution. Moreover, training for the occupancy network is simple without using any additional information that other methods do not have access to (see A3 for more details).
>
> (2) The occupancy field collaborates well with the keypoint field. Although the saliency of an arbitrary point can be obtained through querying the feature point field, the occupancy of the point also needs to be known to further filter the points that lie on the input's surface. Compared with the point cloud, the occupancy field can easily tell us the geometric information of any query point.
>
> ---
>
> **Q3**:
>
> > The authors claim that the proposed method is a novel unsupervised method. This is also not correct, since this method requires ground truth occupancy information as a supervision.
>
> **A3**:
>
> Sorry for this misunderstanding.
>
> (1) Unsupervised means the keypoint location must be learned in an unsupervised manner (like USIP and UKPGAN) because it is ambiguous for humans to label keypoints.
>
> (2) Meanwhile, we do not use ground truth occupancy values. We only use input point clouds to learn this implicit occupancy function. We randomly sample the positives from the input point cloud. Moreover, the negatives are randomly sampled in the unit 3D space. Although some of the negatives are indeed on the surface of the object, their number is so limited compared to the whole query sets that they do not affect the training. Therefore, we do not use any additional information that USIP and UKPGAN do not use. We have added these notes in the training details (see section B.1) in the revised supplementary.

---

> > ### Comment · Reviewer_xHwu · 2022-08-03
> > **Some replies are not convincing**
> >
> > Thank the authors for their efforts in the rebuttal. I read through all the reviews from all reviewers and all your replies in this rebuttal. Some of your replies are helpful to make your paper clearer to me. Unfortunately, some of your replies are not convincing at all.
> >
> > **1. The proposed method is not unsupervised. It needs ground truth occupancy information as an additional supervision, which results in unfair comparisons with other methods.**
> >
> > As the authors claimed many times, the proposed method is unsupervised, while this is not true. As shown by the loss Lo in Eq.(4), it requires occupancy supervision to learn an implicit function for surface reconstruction. This is much different from the self-reconstruction for point clouds, as illustrated by UKPGAN in Fig.1(b), since the input point clouds are also the reconstruction target, and no additional supervision is required.
> >
> > The authors replied to me in A3.(2) in the 3/3 section, they merely obtained the occupancy information by sampling positives from the input point cloud and negatives from the unit 3D space. But you definitely have to determine the area that is not occupied and the area that is occupied. If you did not know this information, you would have lots of queries sampled inside of the shape while labelling them as negative.
> >
> > **2. It makes no sense to use occupancy probability to represent the inverse distances between the query and the input.**
> >
> > The authors replied in A.7 in the 2/3 section, they said that “in the modern literature on occupancy, the shape is not consider solid”. This is not common sense about the occupancy. To learn a valid implicit function, we regard the area inside of the shape as the occupied and outside of the shape as the unoccupied, no matter if the shape is a voxel grid or mesh. It seems there is no one merely regarding the surface as the occupied while other areas are left as the unoccupied. If the authors used the occupancy in this way, I do not think they can reconstruct surfaces like the ones shown in paper.
> >
> > If the shape is regarded as an occupied solid, it makes no sense to model the inverse distance between the query and the input using the occupancy probability. Since the changing patterns of occupancy probability and inverse distances across the surface are so different.
> >
> >
> > **3. It is not conclusive that the occupancy information can improve the keypoint detection accuracy in Table.2.**
> > The results without occ and with all in the second column indicates that the surface reconstruction cannot improve the keypoint detection accuracy under ModelNet40. There should be no reason to obtain results like this if the reconstruction really helps. I can not accept the authors’ explanation in A11 in section 3/3.
> >
> > **4. Novelty and contributions**
> > - Although the authors proposed some loss terms to make them work for keypoint detection, I do not think the novelty is high enough compared to the UKPGAN. The idea of using shape reconstruction to improve the keypoint detection is the same.
> >
> > - Moreover, I do not think it is a wise solution to detect keypoints from a continuous field defined in the whole 3D space since we know keypoints should appear on the surface represented by the input point clouds.
> >
> > - Surface reconstruction from point clouds without normals is a much harder problem than keypoint detection, which is still a challenge. Detecting keypoints based on surface reconstruction makes the problem even harder to resolve. I do not think the authors aim to learn an occupancy field that we usually discussed and have a common sense in surface reconstruction. More importantly, it does no improve keypoint detection performance under ModelNet40.

---

> > > ### Author Response · Authors · 2022-08-07
> > > **Responses to Reviewer #xHwu**
> > >
> > > We would like to thank R#xHwu for the detailed feedback. Here we respond to raised concerns one by one.
> > >
> > > **Q1 and Q2**
> > > > 1. The proposed method is not unsupervised.
> > >
> > > > 2. It makes no sense to use occupancy probability to represent the inverse distances between the query and the input.
> > >
> > > **A1 and A2**
> > >
> > > Sorry for the insufficient description of the occupancy we predict in this paper, which is indeed surface occupancy instead of shape occupancy.
> > >
> > > Conventionally, points inside or on the input surface are considered occupied, as you stated. One needs ground-truth occupancy information (e.g., generated from CAD models) to learn an implicit occupancy function under this formulation.
> > >
> > > However, in our formulation, occupied points are those on the input surface, and the others are all considered unoccupied, including the points inside the surface. In order to avoid this ambiguity, we refer to occupancy in the revised paper as surface occupancy (now marked blue).
> > >
> > > Under this surface occupancy definition, the training for the implicit shape field is unsupervised. We randomly sample positives (occupied points) from the input point cloud and negatives (unoccupied points) in the unit 3D space. We visualize the predicted occupancy field in Fig.16 of the revised supplementary. These five samples are taken from the unseen test set. As shown by the second row, only points on the input surface have a high occupancy value, and the other points (inside or outside of the surface) have a near-zero occupancy value. Under our formulation, two surfaces can be obtained through the marching cube algorithm (using a threshold of 0.4), and we only show the outer surface. It also could be found in Fig.16, the (surface) occupancy probability can represent the inverse distances between the query and the input surface.
> > >
> > > Because we do not use any additional information that other methods do not have access to, the comparison is fair. Finally, since you insist that our formulation is not possible, we recommend you try our codes which clearly show that it is indeed possible.
> > >
> > > **Q3**:
> > > > It is not conclusive that the occupancy information can improve the keypoint detection accuracy in Table.2.
> > >
> > > **A3**:
> > >
> > > (1) As you stated, keypoints should be on the input surface. Without the surface occupancy decoder, keypoints cannot be constrained on the input surface so that some of them would float in the air, as seen in Fig.7-(a). If the detector has a surface occupancy decoder, it could encourage the keypoint to be located on the underlying surface, with the help of proposed loss functions.
> > >
> > > (2) To reconstruct the shape across instances of the same category, our model naturally encourages semantic consistency of the intermediate feature embedding. So the keypoints detected could be semantically consistent. Without the shape decoder, the model performs poorly on the KeypointNet dataset.
> > >
> > > (3) We would like to clarify that SE(3) repeatability is important but not equivalent to accuracy. Our evaluation and analysis clearly show that SE(3) repeatability has its own limitations. And it is clearly shown that, apart from this important but imperfect metric, our method generates meaningful keypoints consistent with human annotation and achieves better registration performance.
> > >
> > > **Q4**
> > > > Novelty and contributions.
> > >
> > > **A4**:
> > >
> > > (1) UKPGAN reconstructs input point cloud coordinates instead of the underlying shape manifold. Moreover, it can only predict keypoints from the input point cloud, which cannot maintain a consistent performance when the test point clouds are disturbed. We propose the first implicit keypoint field that is tightly combined with the implicit surface occupancy field, which is novel.
> > >
> > > (2) Input point clouds are not equivalent to the input surface. A detector should extract consistent keypoints under a number of disturbances that can affect the input point clouds, e.g., missing parts, point density, and sensor noise. A continuous keypoint field that can be queried at any point in 3D space is more consistent under input disturbances.
> > >
> > > (3) We respectfully disagree with your point that shape reconstruction is harder than 3D keypoint detection. It is ambiguous for humans to label keypoints, so learning-based keypoint detection must be done in an unsupervised manner. Meanwhile, shape reconstruction is definitely another hard task and incorporating it helps the continuous keypoint field better capture shape cues.

---

> ### Author Response · Authors · 2022-08-02
> **Responses to Reviewer #xHwu 2/3**
>
> **Q4**:
>
> > The proposed method is also not novel. UKPGAN has explored the feasibility of combining shape reconstruction and keypoint detection together, although UKPGAN reconstructs the input point cloud rather than learning an implicit function to represent the same shape. The minor difference here is what representation is used to represent 3D shapes.
>
> **A4**:
>
> Thank R#xHwu for the comment, but we cannot fully agree with the comment.
>
> (1) We are the first to propose a keypoint field that predicts a keypoint saliency value for each continuous input query point coordinate. The advantages of the keypoint field are stated in A1.
>
> (2) We propose several novel loss functions that exploit the mutual relationship between two keypoint and occupancy decoders, which is quite different from UKPGAN.
>
> (3) We design a gradient-based optimization strategy for refining the keypoint localization during inference.
>
> ---
>
> **Q5**:
>
> > Learning occupancy field results in using an additional surface constraint. This term should not be there due to the learning of the occupancy field. Fig.2(b) may be confusing, I think the close to 0 line should be parallel to the x axis rather than y axis.
>
> **A5**:
>
> The middle panel of Fig.2 indicates the loss functions for keypoint field learning. Surface constraint loss, which entangles the occupancy and keypoint fields, enforces the saliency of the query that is far from input close to 0 (see Eq.6 in the main paper). Since it plays a vital role in formulating the keypoint field, it should appear there.
>
> Thank R#xHwu for the suggestion. We rotate 'close to 0 line' in the revised main paper.
>
> ---
>
> **Q6**:
>
> > The term of repeatability loss is hard to understand. Why do we need a term like this? For shapes, they are well aligned, why do we need to consider the rigid transformation? Another question is about the cosine similarity, the probabilities are scalars, right? Is there any reason to use cosine similarity to evaluate the difference between two numbers?
>
> **A6**:
>
> Repeatability means we need to detect the same keypoints under various transformations so that later geometric tasks can be successfully done (like registration), so applying SE(3) transformations is a natural choice. Enforcing repeatability under SE(3) transformations allows us to detect keypoints in an unsupervised manner (i.e., without human annotated keypoints for supervision).
>
> Yes, for a single point, the value is a scaler. But we calculate the cosine similarity between vectorized values in a local grid so that contextual information that reflects local shape can be captured. (see lines 140-149 in the main paper)
>
> ---
>
> **Q7**:
>
> > The authors use occupancy probability to represent the inverse distance between the query and the input. I do not think it works here. Since occupancy probability always gets smaller from 1 to 0 when moving a query from inside to outside while the inverse distance gets smaller to larger before going across the surface, and then gets smaller after that.
>
> **A7**:
>
> In the modern literature on occupancy, the shape is not considered solid. Again, note that we do not use the solid CAD model for ground truth occupancy calculation. All we need is the input point cloud for training. See A3 for details.
>
> ---
>
> **Q8**:
>
> > Explicit keypoint extraction via optimization is also an operation that I do not understand. An intuitive idea of extracting keypoints is to use points of the input point clouds as queries to predict the occupancy values since keypoints can only locate on the surface. Why do we have to extract keypoints via updating the locations of randomly sampled queries in the 3D space?
>
> **A8**:
>
> No, we do not optimize from randomly sampled points. At inference, the initial query set is evenly distributed in the input space, and then there is a surface filtering step (see Algorithm 1 and Fig.2-inference in the main paper). This step refines keypoint locations without moving queries into the void because the local maxima of the saliency field lie on the input surface. If the initial query set is the input point cloud, its number and distribution will be affected by the input.
>
> ---
>
> **Q9**:
> > In the visualization of saliency field slices, how to select the slices to visualize?
>
> **A9**:
>
> Sorry for this misunderstanding. We implement this visualization by projecting the keypoint field onto an axis (i.e., using torch.max()). We have updated the paper and renamed it as the 'projected slice'.

---

> ### Author Response · Authors · 2022-08-02
> **Responses to Reviewer #xHwu 3/3**
>
> **Q10**:
> > How to determine the keypoint number? I noticed that the numbers of keypoints produced by different methods are different even in the same case, such as the visual comparison in Fig.6. Why do not use the same number of points to perform the comparison？
>
> **A10**:
>
> Sorry for the insufficient description of the visualization. For all quantitative experiments, we fixed the number of keypoints detected by each method for a fair comparison. However, the visualized keypoints in Fig. 6 are selected by Non-Maximum Suppression with a radius of 0.1, similar to how the keypoints are demonstrated in USIP. Therefore, the different methods may show a different number of keypoints of the same object. Notably, ISS cannot predict keypoint saliency, so we randomly choose 30 points from all predicted keypoints for an object and 100 points for an indoor scene. In the supplementary, more qualitative results can be found in Fig.7-Fig.15.
>
> ---
>
> **Q11**:
>
> > The reason why I do not think the reconstruction can improve the performance of keypoint detection is the results in Table 2. The results without surface reconstruction are better than the results with surface reconstruction. I do not think this is a good support for the argument.
>
> **A11**:
>
> As we stated in lines 262-267 of the main paper, although the model without shape reconstruction could detect more repeatable keypoints on the ModelNet40 dataset, it fails to give semantically consistent keypoints on the KeypointNet dataset. Fig. 7-a in the main paper shows that SNAKE is unable to output symmetric and meaningful keypoints without the shape-aware technique.
>
> ---
>
> **Q12**:
>
> > With the ground truth occupancy supervision, it is not a fair comparison with other methods.
>
> **A12**:
>
> No, we do not use ground truth occupancy. We do not use any additional information that other methods do not have access to, so the comparison is fair. The only supervision signal we use is the input point cloud itself. See A3 for more details.

---

### Official Review · Reviewer_2TwX · 2022-07-12

**Rating:** 5
**Confidence:** 3
**Soundness:** 3 good
**Presentation:** 4 excellent
**Contribution:** 3 good

**Summary:**

The paper presents an unsupervised method to predict 3d keypoints from point cloud. Several novel losses are proposed to enforce repeatability, Surface Constraint, and Sparsity. They also achieve superior performance on various public benchmarks. They also use the novel two head to model occupancy and saliency separately to better disentangle these two tasks and let them serve as independently different functions.

**Questions:**

Are there any other newer methods to compare with? The UKPGAN seems to be in the year 2020.
Can you elaborate more on discrete space and continuous space?
Can you elaborate more of the difference when compared with "R2d2: Reliable and repeatable detector and descriptor"?

**Ethics Review Area:**

["I don’t know"]

**Limitations:**

Yes

**Strengths And Weaknesses:**

Strengths：
I like the intuition that starts from continuous instead of discrete space.
The proposed architectures and losses are novel as far as I see.
The presentation is very well.
It achieves the SOTA performance on several datasets.
Weaknesses：
Lack of related work discussion: FULLY CONVOLUTIONAL MESH AUTOENCODER USING EFFICIENT SPATIALLY VARYING KERNELS

---

> ### Author Response · Authors · 2022-08-02
> **Responses to Reviewer #2TwX**
>
> We would like to thank R#2TwX for the professional assessment. Here we respond to raised concerns one by one.
>
> ---
>
> **Weaknesses**:
>
> > Lack of related work discussion: FULLY CONVOLUTIONAL MESH AUTOENCODER USING EFFICIENT SPATIALLY VARYING KERNELS
>
> **A**:
>
> We thank R#2TwX for suggesting this highly related paper. We have referred to it as [43] (line 110) in the revised main paper. We believe it to be applicable, after some adaptation, to our framework as a shape encoding/reconstruction component. For now, the major obstacle to use it is that an additional meshing step is needed since our input is a point cloud without mesh connectivity. We think this can be left for future work.
>
> ---
>
> **Q1**:
>
> > Are there any other newer methods to compare with? The UKPGAN seems to be in the year 2020.
>
> **A1**:
>
> To clarify, the UKPGAN paper firstly appeared on Arxiv in 2020, but it was finally accepted to CVPR 2022. This fact can be checked on this link:
>
> https://openaccess.thecvf.com/content/CVPR2022/html/You_UKPGAN_A_General_Self-Supervised_Keypoint_Detector_CVPR_2022_paper.html
>
> During 2020-2022, this paper underwent substantial revision, and we refer to its latest version. We believe a CVPR 2022 paper can be considered a state-of-the-art and reasonable baseline. To avoid this kind of confusion, we have updated the reference in the main paper to the CVPR 2022 version.
>
> ---
>
>
> **Q2**:
>
> > Can you elaborate more of the difference when compared with "R2d2: Reliable and repeatable detector and descriptor"?
>
> **A2**:
>
> In our opinion, the differences between SNAKE and R2D2 include:
>
> (1) Since R2d2 predicts saliency scores for a 2D image, the keypoint location comes from discrete grids of pixels. By contrast, we use the coordinate-based networks, which parameterize keypoint probability as a continuous function.
>
> (2) Our method tightly entangles the shape reconstruction and keypoint detection, which brings several advantages. While R2d2 does not introduce a task for image reconstruction.
>
> (3) Because 3D keypoints are encouraged to lie on the surface of the input, we propose a novel surface constraint loss that utilizes the occupancy probability and keypoint saliency. In contrast, since R2d2 detects keypoints from the 2D image plane, R2d2 does not propose a loss function for similar mutual constraint purposes.
>
> (4) We can further refine the coordinates of keypoints by gradient-based optimization in the continuous keypoint field, which is also different from R2d2.
>
> ---
>
> **Q3**:
>
> > Can you elaborate more on discrete space and continuous space?
>
> **A3**:
>
> Traditional signal representations are discrete - for example, 2D images are discrete grids of pixels, and 3D data are often represented as meshes, voxels, or point clouds. While implicit neural representation aims to parameterize a signal as a *continuous function* by a neural network that outputs whatever is at a given coordinate, for example, occupancy, SDF, and radiance.
>
> This paper focuses on 3D keypoint detection from point clouds. The former work UKPGAN estimates the saliency of each point in the input point clouds and selects the most salient point as keypoints. However, discrete point clouds use a finite number of points to represent the 3D object or scene. It's possible that keypoints selected from this discrete set are sub-optimal due to limited sampling of the input. So we propose a keypoint field that is a continuous representation. We can predict the saliency of points at arbitrary continuous coordinates. When the input point clouds are down-sampled or affected by noises, our method outperforms counterparts that rely on discrete keypoint representations.

---

> ### Author Response · Authors · 2022-08-08
> **Responses to Reviewer #2TwX**
>
> Dear reviewer,
>
> Please let us know if our responses have addressed the issues raised in your review. We hope that our corrections, clarifications, and additional results address the concerns you've raised. We are happy to address any further concerns.

---

### Official Review · Reviewer_5p2u · 2022-07-14

**Rating:** 6
**Confidence:** 5
**Soundness:** 4 excellent
**Presentation:** 3 good
**Contribution:** 3 good

**Summary:**

The paper is well written, the motivation and technical details are clearly presented. The idea of estimating saliency field from sparse keypoints seem novel, and it is shown to be effective to produce repeatable and consistent keypoint detection result.

**Questions:**

Additional visualization of repeatibility of keypoints under SE3 transformation as well as different sparsity of points can make the contribution of the paper more clear. The plots in Figure 5 alone is not visual enough to show the quality of the proposed method in terms of repeatibility.

**Ethics Review Area:**

["I don’t know"]

**Limitations:**

Keypoint extraction during inference requires considerable computational cost, not suitable for real-time on-device applications.

**Strengths And Weaknesses:**

Strength:

The paper is well written, the motivation and technical details are clearly presented. The idea of estimating saliency field from sparse keypoints seem novel, and it is shown to be effective to produce repeatable and consistent keypoint detection result.

Weakness:

Keypoint extraction in inference time requires iterative gradient descent and query the implicitly defined saliency field. Thus the computational cost is high and is not suitable for real-world applications at its current form.

---

> ### Author Response · Authors · 2022-08-02
> **Responses to Reviewer #5p2u**
>
> We would like to thank R#5p2u for the professional assessment. Here we respond to raised concerns one by one.
>
> ---
>
> **Weakness**:
>
> > Keypoint extraction in inference time requires iterative gradient descent and query the implicitly defined saliency field. Thus the computational cost is high and is not suitable for real-world applications at its current form.
>
> **A**:
>
> Thanks for pointing out this. We would like to note that the gradient descent optimization step is an optional add-on, which trades off computational overhead for higher accuracy. Without this step, our method can still achieve strong keypoint detection results, as evidenced by:
>
> (1) Figure 7-c in the main paper shows that when no optimization is used (0-step), the repeatability is still as high as around 81%.
>
> (2) Table 4 in the supplementary material shows that when no optimization is used, our method is as fast as the traditional method ISS. Note that ISS is widely used in online robotics applications like [1][2].
>
> In addition, SNAKE requires the lowest GPU memory cost to generate keypoints compared with other deep learning based methods, as shown by Figure 2 in the supplementary material. It shows that SNAKE is not only efficient in terms of speed but also memory usage.
>
> Finally, to better illustrate the trade-off between speed and accuracy, we have added keypoint repeatability into Table 4 of the supplementary material, which is also shown below.
>
> Table 1: Average time (s) taken to compute keypoints from input point clouds on ModelNet40 dataset. Decimals in parentheses in italics are relative repeatability (%). Here, the experiment setting is the same as section 4.2 in the main paper. $J$ is the optimization step.
>
> | Input Point # |      ISS       |      USIP       |   Ours $J$=0   |   Ours $J$=5   |  Ours $J$=10   |
> | :-----------: | :------------: | :-------------: | :------------: | :------------: | :------------: |
> |     2048      | 0.07 (*0.088*) | 0.006 (*0.748*) | 0.08 (*0.795*) | 0.50 (*0.835*) | 0.81 (*0.851*) |
> |     4096      | 0.11 (*0.096*) | 0.007 (*0.799*) | 0.09 (*0.811*) | 0.50 (*0.850*) | 0.83 (*0.864*) |
>
>
> [1] F. Fadri, et al. Autonomous robotic stone stacking with online next best object target pose planning. In IEEE international conference on robotics and automation (ICRA), pages: 2350-2356, 2017.
>
> [2] Jiadong Guo, et al. Local Descriptor for Robust Place Recognition Using LiDAR Intensity. IEEE Robotics and Automation Letters, 4(2):1470–1477, 2019.
>
> ---
>
> **Q1**:
>
> > Additional visualization of repeatability of keypoints under SE3 transformation as well as different sparsity of points can make the contribution of the paper more clear. The plots in Figure 5 alone is not visual enough to show the quality of the proposed method in terms of repeatability.
>
> **A1**:
>
> We thank R#5p2u for this suggestion. We have added some new qualitative visualization figures (Figure 13-15) in the supplementary material, which compare the repeatability of different methods using the same randomly generated SE(3) transformation.

---

> ### Author Response · Authors · 2022-08-08
> **Responses to Reviewer #5p2u**
>
> Dear reviewer,
>
> Please let us know if our responses have addressed the issues raised in your review. We hope that our corrections, clarifications, and additional results address the concerns you've raised. We are happy to address any further concerns.

---

### Official Review · Reviewer_ZdeR · 2022-07-15

**Rating:** 6
**Confidence:** 4
**Soundness:** 3 good
**Presentation:** 3 good
**Contribution:** 3 good

**Summary:**

The paper presents a novel unsupervised method SNAKE to detect 3D keypoints from point clouds based on implicit neural representations. The key idea is to combine shape reconstruction and keypoint detection during training. Experiments show that jointly learning 3D shapes and key points improves semantical consistency, better repeatability under disturbance, and accurate geometric registration under zero-shot settings.


**Questions:**

- L224 states that UKPGAN is not involved in the experiments due to the absence of pretrained model. Since their code is publicly available, I wonder if training from scratch is possible? If not, I wonder if it is possible to compare alternative datasets?
- The registration experiments (sec 4.3) rely on D3Feat descriptors for the detected keypoints. I am aware that this descriptor is commonly used in the UKPGAN’s experiment. I am interested in understanding the bottleneck of the problem. Since D3Feat detector + D3Feat descriptor still serves as an upper bound in this experiment, I wonder whether it is possible that certain detector requires specifically designed descriptors in order to work perfectly in the registration problem. D3Feat descriptor may not provide the best features for SNAKE or UKPGAN.


**Limitations:**

Yes.

**Strengths And Weaknesses:**

+ Though the idea of combining reconstruction and saliency prediction is not new (UKPGAN also reconstructs shape), this paper takes advantage of implicit representation and shows advantages over the GAN-based method. The proposed method is simple but effective. The proposed four loss functions are intuitive and ablations show that they are important.
+ The experiments are mostly thorough and show good qualitative and quantitative results compared to competitive baselines.
+ The paper is well written and the figures are easy to understand.
- Missing comparison with UKPGAN in section 4.2. UKPGAN is a competitive baseline and its code is publicly available.

Post-rebuttal:
My final rating is weak accept. Thanks to the authors and reviewers for their effort. The rebuttal mostly answers my questions. I think the paper is novel and has enough difference from UKPGAN -- the method does not have a GAN and the experiment results are better. However, I do think D3Feat descriptor may not be the best way to evaluate the proposed method in experiments since it is designed for D3Feat detector and may not be discriminative enough for different feature detectors. Since the prior works are using this protocol, I won't criticize too much for following it.

---

> ### Author Response · Authors · 2022-08-02
> **Responses to Reviewer #ZdeR 1/2**
>
> We would like to thank R#ZdeR for the professional assessment. Here we respond to raised concerns one by one.
>
> ---
>
> **Weakness and Q1**:
> > Missing comparison with UKPGAN in section 4.2. UKPGAN is a competitive baseline and its code is publicly available.
> >
> > L224 states that UKPGAN is not involved in the experiments due to the absence of pretrained model. Since their code is publicly available, I wonder if training from scratch is possible? If not, I wonder if it is possible to compare alternative datasets?
>
> **A1**:
>
> We thank R#ZdeR for this suggestion. We have tried to train UKPGAN (official implementation) on the ModelNet40 and 3DMatch datasets from scratch but observed divergence under default hyper-parameters. The training always reports NaN losses in early epochs. This instability also implies limitations in implementing the idea of joint reconstruction and keypoint detection with GAN-based methods.
>
> However, we do agree that comparing repeatability (apart from semantic consistency and registration accuracy) between UKPGAN and SNAKE is necessary. As such, we provide a new experiment to compare their repeatability on the KeypointNet dataset, on which the UKPGAN provided a pre-trained model. We randomly perform SE(3) transformation on the test point clouds to generate the second view point clouds. Then, we select top-32 salient keypoints with NMS (radius=0.03) in each sample and show the keypoint repeatability under different distance thresholds $\epsilon$, downsample rates, and Gaussian noise scales. The relative repeatability (%) results are summarized as follows:
>
> Table 1:  Relative repeatability (%) with different distance thresholds $\epsilon$ on the KeypointNet dataset.
>
> | $\epsilon$ |   0.03    |   0.04    |   0.05    |   0.06    |   0.07    |   0.08    |   0.09    |   0.10    |
> | :--------: | :-------: | :-------: | :-------: | :-------: | :-------: | :-------: | :-------: | :-------: |
> |   UKPGAN   |   0.199   |   0.322   |   0.454   |   0.564   |   0.661   |   0.741   |   0.810   |   0.864   |
> |    Ours    | **0.643** | **0.734** | **0.806** | **0.856** | **0.892** | **0.918** | **0.936** | **0.948** |
>
> Table 2:  Relative repeatability (%) when input point clouds are disturbed ($\epsilon$=0.03) on the KeypointNet dataset.
>
> |        | Original  |  Down 4x  |  Down 8x  | Noise std=0.02 | Noise std=0.03 |
> | :----: | :-------: | :-------: | :-------: | :------------: | :------------: |
> | UKPGAN |   0.199   |   0.570   |   0.427   |     0.608      |   **0.558**    |
> |  Ours  | **0.643** | **0.594** | **0.525** |   **0.626**    |     0.536      |
>
> These two tables show that SNAKE achieves significant gains over UKPGAN in most cases. Interestingly, when the inputs are disturbed, the performance of UKPGAN increases rather than decreases. Via visualizing the results (see Fig.1 of the updated supplementary material), we find that when the input point clouds are disturbed, the keypoints predicted by UKPGAN are clustered in a small area, which improves the repeatability of keypoints but fails to cover the input uniformly. This illustrates that the GAN-based method adopted by UKPGAN to control the keypoint sparsity is not robust to input point cloud disturbance. The keypoints of ours still remain meaningful under the drastic changes of inputs.
> These results are also updated in the revised supplementary material, as can be found in section C.1.

---

> > ### Author Response · Authors · 2022-08-02
> > **Responses to Reviewer #ZdeR 2/2**
> >
> > **Q2**:
> >
> > > The registration experiments (sec 4.3) rely on D3Feat descriptors for the detected keypoints. I am aware that this descriptor is commonly used in the UKPGAN’s experiment. I am interested in understanding the bottleneck of the problem. Since D3Feat detector + D3Feat descriptor still serves as an upper bound in this experiment, I wonder whether it is possible that certain detector requires specifically designed descriptors in order to work perfectly in the registration problem. D3Feat descriptor may not provide the best features for SNAKE or UKPGAN.
> >
> > **A2**:
> >
> > (1) Firstly, we would like to note a fact: the D3Feat detector + D3Feat descriptor is hard to surpass but not a strict upper bound. As demonstrated by Table.1 in the main paper, SNAKE + D3Feat descriptor outperforms D3Feat detector + D3Feat descriptor under 1000/500 points for feature matching recall, 2500 points for registration recall, and 2500 points for inlier ratio. This fact indicates that when the number of keypoints is high enough, SNAKE can collaborate well with off-the-shelf descriptors.
> >
> > As such, if it is allowed to use many keypoints (e.g., 2500), the performance bottleneck actually lies in the registration problem itself (e.g., local minima under a certain SE(3) transformation).
> >
> > (2) Secondly, if only a small number of keypoints are allowed, the mismatch between detector and descriptor is the bottleneck. For example, by the shape-aware mechanism, SNAKE could consider the center of a flat table top to be a keypoint as it reflects the geometric center. In contrast, the descriptors of D3Feat near the center of a flat table top may not be discriminative enough for registration.

---

### Comment · Area_Chair_NuGy · 2022-08-08
**Any thoughts from reviewers?**

Hi Reviewers,

The discussion period is closing soon. Please take a look at the responses from the authors. If you have further questions, please ask them now, since the authors will be unable to respond soon. It's substantially more productive, effective, and reasonable to have a quick back-and-forth with authors now than to raise additional questions or concerns post-discussion period that the authors are unable to address.

Thanks,

AC

---

### Meta-Review · Area_Chair_NuGy · 2022-08-24

**Recommendation:** Accept
**Confidence:** Certain

**Metareview:**

Post-rebuttal, the paper had split reviews, with three reviewers in favor of acceptance (6, 6, 5 but noted as a 6 in the final comment from 2TwX) and one reviewer strongly arguing for rejection (3). The AC examined the reviews, the paper, and the discussion, and is inclined to accept the paper. The AC is persuaded by the arguments presented by the reviewers in favor of acceptance. While xHwu has raised a number of concerns, the AC believes that the authors have addressed most of these well in public discussion. The AC understands xHwu's positions, but does not see the remaining concerns as grounds for rejection in light of the more positive views of the other reviewers. Given the extensiveness of the discussion, the AC would encourage the authors to use their extra page to incorporate some of the experimental results into the final version of the paper.

**Award:**

No

---

### Decision · Program_Chairs · 2022-09-14

Accept